# Effect of Heat Input on Microstructure and Corrosion Resistance of CP-Ti Laser Beam Welded Joints

**Zhen Li** [1,2]**, Wei Zhao** [1,2,*]**, Kedong Yu** [1,2]**, Ning Guo** [1,2] **and Song Gao** [1,2]

1  College of Mechanical Engineering, Qilu University of Technology (Shandong Academy of Sciences), Jinan 250353, China
2  Shandong Institute of Mechanical Design and Research, Jinan 250031, China
*  Correspondence: zwapple@yeah.net

**Abstract:** The TA1 welded joints with different heat inputs were obtained by a fiber laser and their microstructure, mechanical properties and corrosion resistance in simulated saliva solution were studied. The results show that the microstructure in fusion zone (FZ) is needle-like $\alpha'$ martensite and lath-shape $\alpha'$ martensite, and that of the heat-affected zone (HAZ) is zigzag $\alpha$ phase. With the increase of heat input, the volume fraction of needle-like $\alpha'$ martensite decrease and the microstructure is coarsened in FZ, but there is almost no change in the microstructure of the HAZ. The order of the corrosion resistance of welded joints with different heat inputs is the same as FZ > HAZ > base material (BM), and the heat input has a more influence on the corrosion resistance of FZ. The binary multiple linear regression relationship between the corrosion current density/charge transfer resistance and the length/width of $\alpha'$ martensite was established, indicating that the width of $\alpha'$ martensite is the main factor affecting the corrosion resistance.

**Keywords:** CP-Ti; laser beam welding; heat input; microstructure; corrosion resistance

## 1. Introduction

In recent years, the problem of oral prosthodontics has attracted more and more attention with the development of material processing technologies. Titanium and its alloys can satisfy a series of requirements for dental materials, such as low Young's modulus and density, superior biocompatibility, high strength-to-weight ratio, and excellent corrosion resistance [1–3]. Among them, commercial pure titanium (CP-Ti) and Ti-6Al-4V (TC4) alloys are the most commonly used. However, Al and V ions of TC4 alloy are easily released during wear or corrosion, which can result in peripheral neuropathy, osteomalacia, Alzheimer's disease, et al. [4–6]. Therefore, the CP-Ti is used as the preferred material for denture restorations, implants, and other applications in many fields of stomatology [7,8].

The production and application of most titanium components depend highly on fusion welding technologies. Titanium and its alloys have low thermal conductivity (22 W cm$^{-1}$ K$^{-1}$) and high affinity with nitrogen, hydrogen, and oxygen at high temperatures, indicating that laser beam welding (LBW) with high energy density and precision is an ideal welding method for them [9,10]. Heat input, as one of the most influential and controllable parameters for LBW, can affect the microstructure and properties of welded joints by changing the flow of the molten pool, cooling rate, geometrical constraints and defect formation. The research on the effect of welding heat input on the microstructure and properties of titanium alloy welded joints was mostly focused on dual-phase titanium alloys (like TC4). Due to the uneven temperature distribution around the melt pool in the welding process, an uneven microstructure is formed in the welded joint (i.e., from the FZ to the BM). In addition, the size and microstructure evaluation in both FZ and HAZ depends on specific heat input [11,12]. With the increase of welding heat input, the $\alpha'$ martensite morphology in FZ changes from needle-like to lath-like and the prior $\beta$ grains are coarsened, deteriorating their mechanical properties [13]. Increasing the heat input in a

certain amount is beneficial to the reduction of the porosity, thereby increasing the tensile strength; while excessive heat input may lead to the formation of welding defects, like undercuts, resulting in the decrease in the tensile strength [14]. In addition, studies have shown that the mechanical properties of welded joints depend on the shape of the weld and heat input is an important factor affecting the shape of the weld [15]. However, there is a tremendous lack of the research on the effects of heat input for CP-Ti laser welded joints.

Corrosion resistance is one of the main considerations for stomatological materials because it not only affects the service life of components but also relates to the harm to living organisms. The corrosion resistance of titanium alloys is related to the phase, grain size, grain boundary density, etc. [16,17]. The complex welding thermal cycle in laser welding process results in the differences in the chemical composition and microstructure of different zones in the welded joints. Many studies have shown that the corrosion resistance of FZ is poorer than the BM in the TC4 welded joint which is mainly attributed to the decomposition of coarse columnar grains and metastable $\alpha'$ martensite [18–21]. However, there are relatively few studies on the corrosion resistance of CP-Ti welded joints and there is a lack of research on the corrosion resistance of welding heat input.

In this study, the effects of heat input on microstructure and of welded joints were characterized by scanning electron microscope (SEM). Mechanical properties of welded joints are evaluated by microhardness tests and tensile tests. In addition, the corrosion resistance of different zones of welded joints was studied by potentiodynamic polarization curve and electrochemical impedance spectroscopy (EIS) in simulated artificial saliva solution that simulating human oral environment, thus exploring the mechanism of corrosion resistance change under different heat input. This work will provide theoretical support for the application of CP-Ti in stomatology.

## 2. Materials and Methods

### 2.1. Materials and Welding Parameters

TA1 plates with a thickness of 4 mm, a width of 50 mm and a length of 60 mm were used. Before laser welding, the plates were dried, polished, and cleaned with ethanol to avoid the interference of external factors on the experimental results. The welding process was carried out by fiber laser (IPG-YLS-10000, IPG Photonics Corporation, Oxford, MA, USA) with a maximum power of 10 kW, an emission wave length of 1070 nm and a spot focus diameter of 0.2 mm. In the welding process, the welding speed was kept constant and the heat input was changed by the laser power under the premise of penetration and qualified forming. The specific parameters are shown in Table 1. In addition, argon gas (99.9% pure) was used as the shielding gas to protect the front and back of the weld, and the flow rates were 20 L/min and 5 L/min, respectively.

**Table 1.** The specific experimental parameters in the laser welding procedures.

| Sample | Power/P (W) | Welding Rate/v (mm/s) | Heat Input/E (J/mm) |
|--------|-------------|------------------------|----------------------|
| S1 | 2000 | 15 | 133.3 |
| S2 | 2400 | 15 | 160 |
| S3 | 2800 | 15 | 186.6 |

### 2.2. Microstructure Observations and Mechanical Properties Testing

After being etched by Kroll's reagent (3% HF + 6% HNO3 + 91% H2O, Tianjin Kermel Chemical Reagent Co., LTD., Tianjin, China), the microstructure of the welded joint was observed by field scanning electron microscopy (FSEM, JEOL-7800F, JEOL Ltd., Tokyo, Japan). According to the SEM results, Image-Pro Plus 6.0 (Media Cybernetics, Rockville, MD, USA) was used to analyze the size of the $\alpha'$ martensite. Before the microhardness test, the sample is polished and polished to ensure that the surface of the sample is flat. Microhardness tests were performed with a microhardness tester (HXD-1000TMC, Xian Weixin Testing Equipment Co., LTD., Xi'an, China) under a loading force of 200 g, a loading

time of 15 s, and the interval between each test point was 0.2 mm. The test range is a rectangular area of 3.8 mm × 8 mm and the center of the rectangle is located at the center of the FZ. The tensile tests were measured by an electronic universal testing machine (Zwick-Z250, ZwickRoell GmbH & Co. KG, Ulm, Germany) and the size of the plate-like tensile specimen was 100 mm × 10 mm × 3 mm, which was repeated three times to ensure the accuracy of the experimental results.

### 2.3. Electrochemical Measurements

The welded joint samples were ground up to 3000 grit silicon, rinsed with deionized water, cleaned ultrasonically with ethyl alcohol and then air dried. During the electrochemical test, the exposed area was 0.1 cm$^2$ and the rest of the sample was covered with insulating glue and silica gel sealed. The experimental device is a conventional three-electrode electrochemical cell with a platinum foil counter electrode and a saturated calomel electrode (SCE) reference electrode and an electrochemical workstation (GAMRY Interface1000, Gamry Instruments Consulting Co., LTD., Shanghai, China). The electrochemical test solution is simulated artificial saliva solution whose composition is shown in Table 2, and the temperature was kept at 37 °C ± 0.1 °C.

**Table 2.** Chemical compositions of simulated artificial saliva solution.

| NaCl | KCl | $CaCl_2 \cdot 2H_2O$ | $Na_2HPO_4 \cdot 2H_2O$ | $Na_2S \cdot 2H_2O$ | $CH_4N_2O$ | Distilled Water |
|---|---|---|---|---|---|---|
| 0.4 g | 0.4 g | 0.795 g | 0.78 g | 0.005 g | 1 g | 1 L |

The $-1.2$ V$_{SCE}$ cathode potential was polarized for 180 s to remove the oxide film at first and then the open circuit potential (OCP) test was performed for 18,000 s. Electrochemical impedance spectroscopy (EIS) was performed at OCP from 0.01 Hz to 100 kHz and the amplitude is 10 mV. The range of the potentiodynamic polarization curve was $-0.5$ to 2 V (vs. OCP) and the scanning speed was 0.5 mV/s.

### 3. Results

### 3.1. Microstructure Evolution

As shown in Figure 1, the microstructure of TA1 is equiaxed $\alpha$ phase whose average grain size is about 54.53 μm.

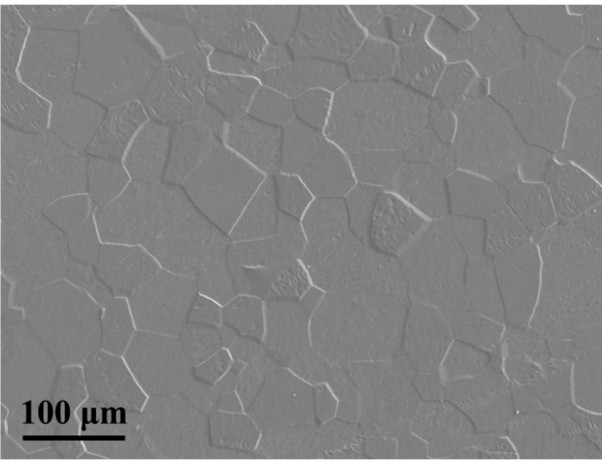

**Figure 1.** Microstructure of the TA1.

The microstructure of different zones in the welded joints with different heat inputs is shown in Figures 2–4. The boundaries of the prior β grains were clearly revealed in the FZ. When the heat input is 133.3 J/mm, long orthogonally oriented martensite plates $\alpha'$ with needle-like morphology and a small amount of lath-shape $\alpha'$ martensite dominates

the microstructure in FZ, and the average length and width of about 8.28 μm and 0.78 μm, respectively. During laser beam welding, the β phase grows along the temperature gradient direction accompanied with the molten pool solidification. The prior β phase is deformed to martensite by shear deformation under a high cooling rate [22]. The microstructure of HAZ is zigzag α phase and its grains are coarser than that of BM (Figure 2c).

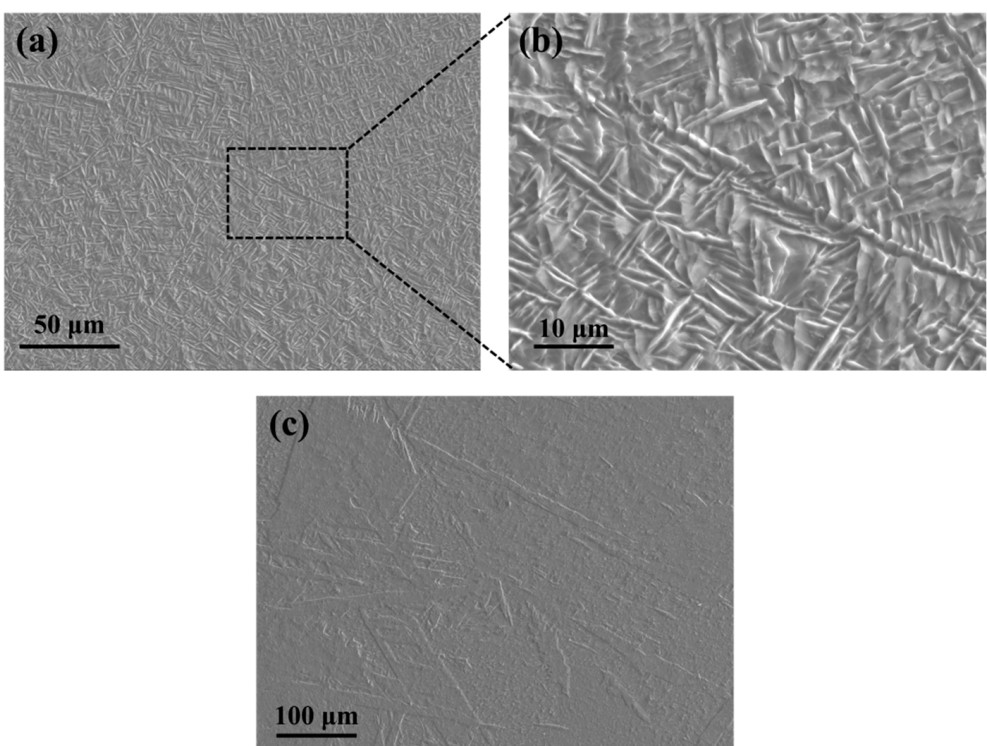

**Figure 2.** Microstructure in different zones of S1: (**a**,**b**) FZ; (**c**) HAZ.

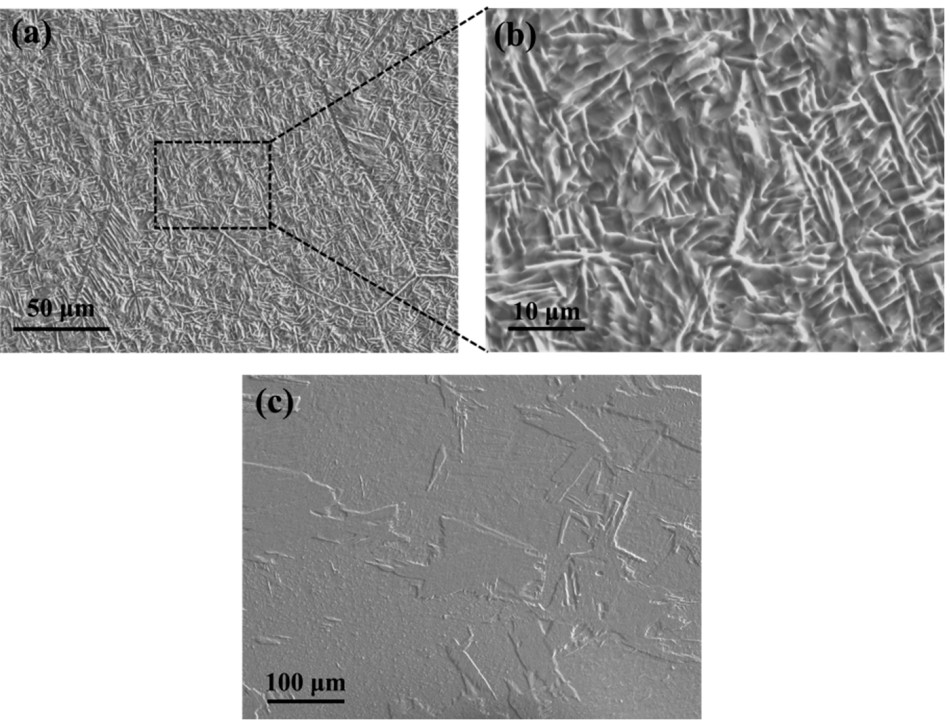

**Figure 3.** Microstructure in different zones of S2: (**a**,**b**) FZ; (**c**) HAZ.

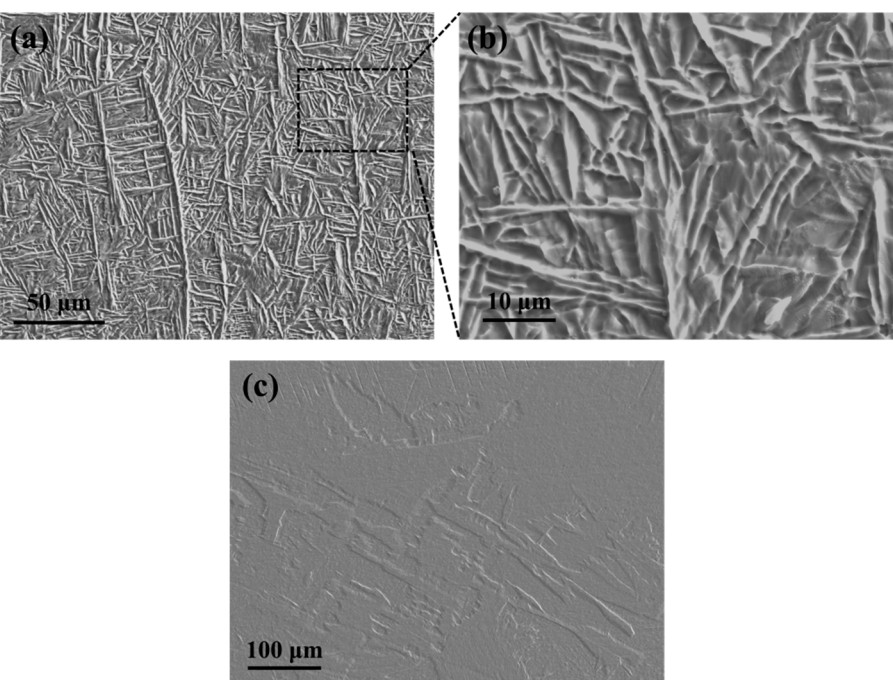

**Figure 4.** Microstructure in different zones of S3: (**a**,**b**) FZ; (**c**) HAZ.

The morphology and grain size of $\alpha'$ martensite in FZ changed with the increase of heat input, i.e., the decrease of cooling rate. As shown in Figure 3a, the volume fraction of needle-like $\alpha'$ martensite decreases and the slat spacing increases when the heat input increases to 160 J/mm. In addition, the grains are coarsened, the width of $\alpha'$ martensite increases slightly, and its average length increases to 9.97 μm. When the heat input increases to 186.6 J/mm, the arrangement of $\alpha'$ martensite was relatively chaotic and the volume fraction of needle-like $\alpha'$ martensite continued to decrease (Figure 4a). The size of $\alpha'$ martensite is significantly coarsened, and its length and width were 1.44 times and 1.46 times bigger than those of the S2 specimen, respectively (Figure 5). However, there is almost no change in the microstructure of the HAZ with different heat inputs.

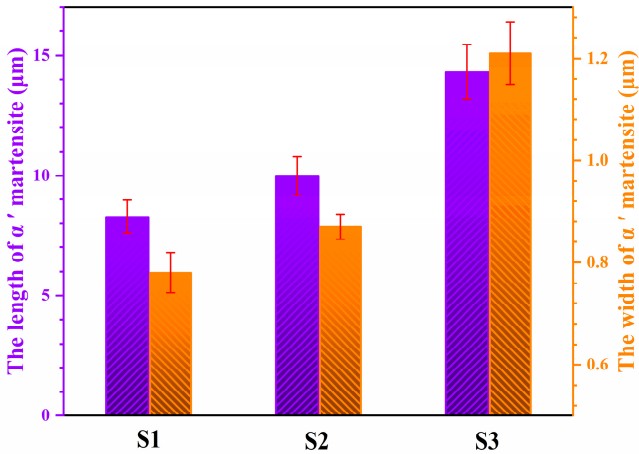

**Figure 5.** Length and width of $\alpha'$ martensite in FZ under different heat inputs.

### 3.2. Mechanical Properties of Welded Joint

Figure 6 shows the microhardness distribution of the welded joints. The three-dimensional image is projected onto the XOY plane indicating the presented shape matches the macroscopic topography of the welded joint. The average microhardness of BM is about 171.2 HV and it decreases gradually from FZ to BM for the welded joints. The highest

microhardness is located in the upper part of FZ and the microhardness of the welded joint gradually decreases with the increase of heat input in which its effect on the upper FZ is the most significant. This may be attributed to the change in cooling rate and the influence of the Marangoni convection effect.

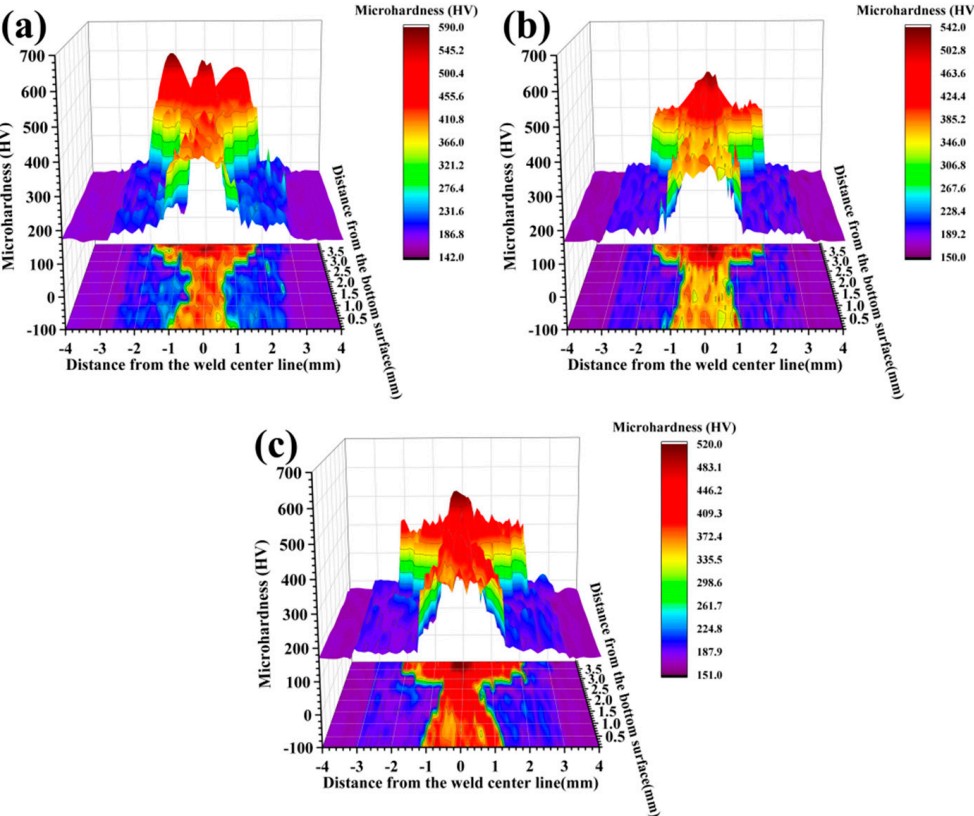

**Figure 6.** Microhardness distribution of welded joints with different heat inputs: (**a**) S1; (**b**) S2; (**c**) S3.

The stress-strain curves of BMs and welded joints with different heat inputs are shown in Figure 7, which can characterize the strength and plasticity of the samples. As shown in Table 3, the strength of the welded joint is bigger and the plasticity is smaller than BM. With the increased of heat input, the yield strength and tensile strength of the welded joint decreased slightly, but the elongation increased slowly.

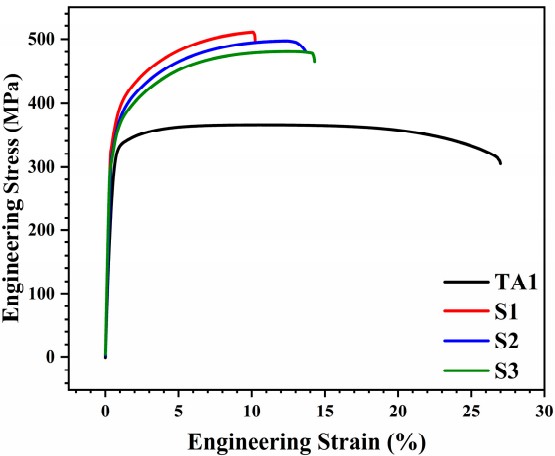

**Figure 7.** Stress-strain curves of TA1welded joints with different heat inputs and the substrate.

**Table 3.** Tensile test results of welded joints.

| Sample | Ultimate Tensile Strength (MPa) | Yield Strength (MPa) | Elongation (%) |
|--------|--------------------------------|---------------------|---------------|
| BM | 365.4 | 298.1 | 26.9 |
| S1 | 510.5 | 343.6 | 10.2 |
| S2 | 497.4 | 341.5 | 13.7 |
| S3 | 480.5 | 317.2 | 14.3 |

*3.3. Corrosion Behaviors*

3.3.1. OCP

As shown in Figure 8, the OCP values of every sample in the welded joints increase continuously and then keep stable which is attributed to the formation of the passive film with the immersion process development. After stabilization, the order of OCP of welded joint is FZ > HAZ > BM ($-364.1$ mV) which indicates that FZ and BM have the best and worst thermodynamic stability, respectively [23]. The change of welding heat input has an obvious effect on the OCP of FZ, and that of S2 is about $-196.4$ mV which is more positive than S1 ($-276.7$ mV) and S3 ($-266.5$ mV).

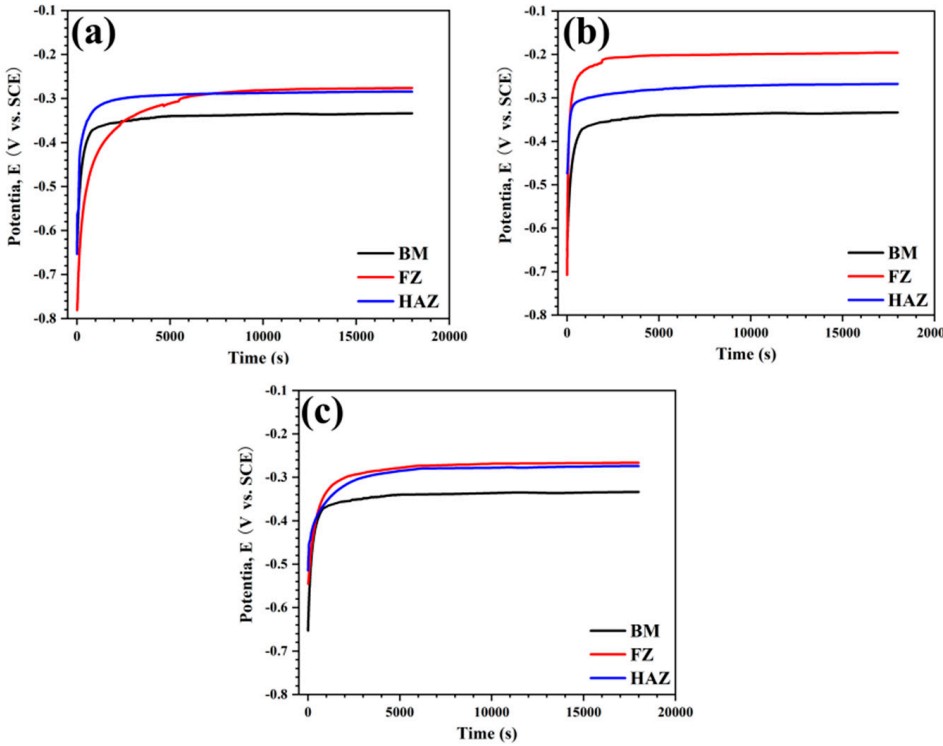

**Figure 8.** OCP diagrams of samples in the welded joints under different welding heat inputs: (**a**) S1; (**b**) S2; (**c**) S3.

3.3.2. Potentiodynamic Polarization

The potentiodynamic polarization curves of samples in the welded joints in the simulated artificial saliva are shown in Figure 9. It can be seen that all samples have similar passivation phenomena and the order of passivation current density ($i_p$) for the welded joints with different heat inputs is BM > HAZ > FZ. Generally, a smaller $i_p$ value represents a more stable passive film [24]. The Tafel extrapolation method is used to fit the corrosion potential ($E_{corr}$) and corrosion current density ($i_{corr}$) of different samples, and the results are shown in Table 4. It can be seen that the law of the $E_{corr}$ is consistent with the OCP. According to Faraday's law, higher $i_{corr}$ represent accelerated corrosion rates [25]. The change of heat input has a great influence on the $i_{corr}$ of FZ, and the $i_{corr}$ of S3 is 1.29 times and 2.77 times than that of S1 and S2, respectively.

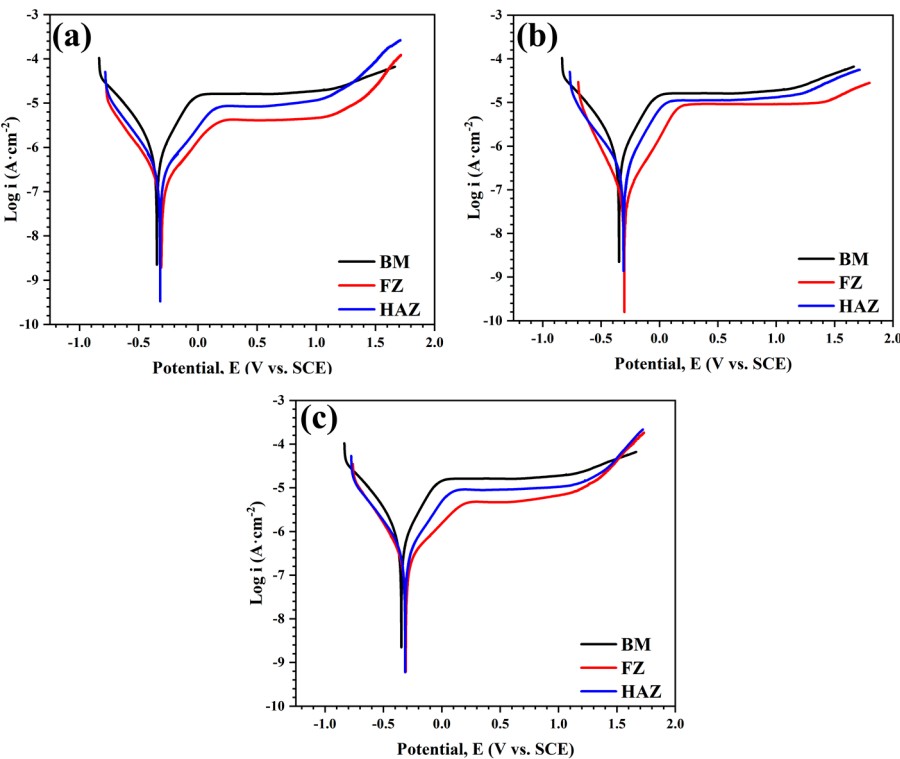

**Figure 9.** Potentiodynamic polarization curves of samples in the welded joint with different heat inputs: (**a**) S1; (**b**) S2; (**c**) S3.

**Table 4.** Fitting results of polarization curves of welded joints.

| | Samples | $E_{corr}$ (mV) | $i_{corr}$ ($\mu A\ cm^{-2}$) | $\beta_a$ (mV dec$^{-1}$) | $-\beta_c$ (mV dec$^{-1}$) |
|---|---|---|---|---|---|
| | BM | −346.2 | 0.649 | 200.5 | 231.7 |
| S1 | FZ | −309.3 | 0.181 | 289.9 | 251.5 |
| | HAZ | −318.6 | 0.318 | 299.3 | 264.6 |
| S2 | FZ | −301.1 | 0.084 | 222.6 | 183.0 |
| | HAZ | −308.7 | 0.303 | 196.9 | 265.2 |
| S3 | FZ | −302.4 | 0.233 | 361.5 | 230.9 |
| | HAZ | −313.8 | 0.339 | 257.7 | 262.8 |

### 3.3.3. EIS

As shown in Figure 10, the Nyquist diagrams of different samples in the welded joint only contain capacitive reactance loops in the first quadrant and the radius of the capacitive loop of BM is the smallest indicating its worst corrosion resistance. In the phase angle part of the Bode diagram, the peak positions of FZ and HAZ move to low frequencies and FZ has a higher phase angle indicating its optimal corrosion resistance. The EIS results were fitted by $R_s(Q_f(R_f(Q_{dl}R_{ct})))$, where $R_s$, $R_f$, and $R_{ct}$ represent solution resistance, passive film resistance, and charge transfer resistance, $Q_f$ and $Q_{dl}$ represent the constant-phase element(CPE) of passive film and electric double layer, respectively. The CPE is used to describe the non-ideal capacitance, and its impedance is given by Equation (1) [26]:

$$Z_{CPE}(w) = \frac{1}{(jw)^n} \qquad (1)$$

where $w$ is the angular frequency, $j$ is the imaginary unit, $n$ is the surface inhomogeneity exponent. The fitting results are shown in Table 5. It can be seen that the $R_f$ and $R_{ct}$ values of FZ and HAZ both increases first and then decrease with the increase of the heat input.

The $R_{ct}$ value of FZ changes more obviously, and those of S2 and S3 are 1.34 times and 0.86 times than that of S1, respectively. The results are consistent with the potentiodynamic polarization curves.

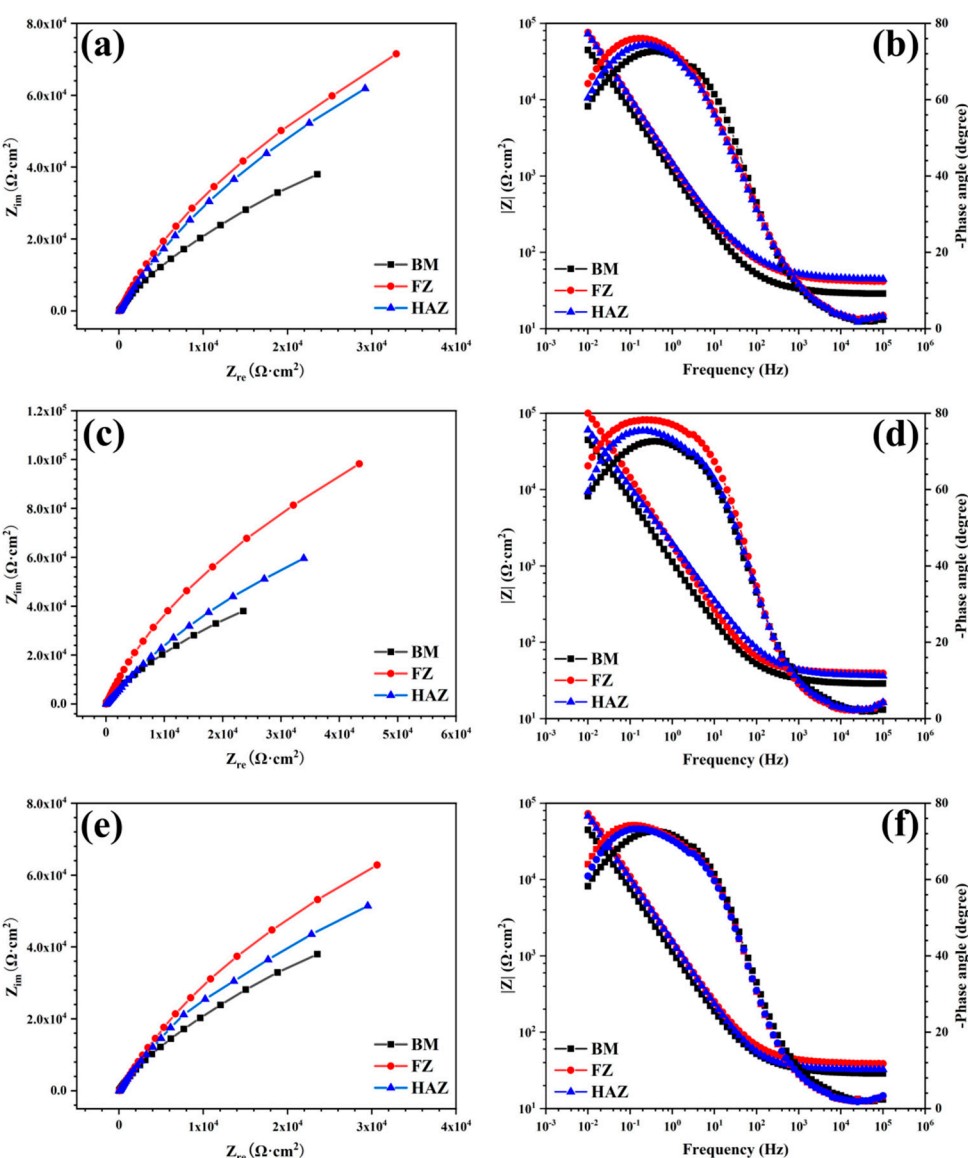

**Figure 10.** EIS diagrams of samples in the welded joint with different heat inputs: (**a,c,e**) Nyquist diagrams; (**b,d,f**) Bode diagrams of S1, S2, S3, respectively.

**Table 5.** The EIS fitting results of welded joint.

| Samples | | $R_s$ ($\Omega \cdot cm^2$) | $Q_f$ | | $R_f$ ($\Omega \cdot cm^{-2}$) | $Q_{dl}$ | | $R_{ct}$ ($\Omega \cdot cm^{-2}$) |
|---|---|---|---|---|---|---|---|---|
| | | | $Y_0$ ($\Omega^{-1} \cdot cm^{-2} \cdot S^n$) | $n_{sl}$ | | $Y_0$ ($\Omega^{-1} \cdot cm^{-2} \cdot S^n$) | $n_{sl}$ | |
| BM | | 29.09 | $9.011 \times 10^{-5}$ | 0.8248 | 35.02 | $1.036 \times 10^{-4}$ | 0.8168 | $1.506 \times 10^5$ |
| S1 | FZ | 43.2 | $6.585 \times 10^{-5}$ | 0.8383 | 133 | $7.394 \times 10^{-5}$ | 0.8689 | $3.652 \times 10^5$ |
| | HAZ | 36.09 | $6.726 \times 10^{-5}$ | 0.8293 | 65.34 | $7.337 \times 10^{-5}$ | 0.8603 | $2.59 \times 10^5$ |
| S2 | FZ | 40.02 | $5.849 \times 10^{-5}$ | 0.8839 | 141.5 | $4.538 \times 10^{-5}$ | 0.8731 | $4.888 \times 10^5$ |
| | HAZ | 39.54 | $7.583 \times 10^{-5}$ | 0.85 | 68.27 | $5.815 \times 10^{-5}$ | 0.8418 | $2.815 \times 10^5$ |
| S3 | FZ | 31.97 | $1.216 \times 10^{-4}$ | 0.8329 | 94.75 | $8.93 \times 10^{-5}$ | 0.8194 | $3.142 \times 10^5$ |
| | HAZ | 34.24 | $8.575 \times 10^{-5}$ | 0.8414 | 46.15 | $4.874 \times 10^{-5}$ | 0.8288 | $2.254 \times 10^5$ |

## 4. Discussion

According to the continuous cooling transformation (CCT) curves, the CP-Ti transformation mode is related to the cooling rate [27]. The β grains transform into α-phase by diffusional in a slow cooling rate [10]. Under moderate cooling rate, the bulk transition occurs which is attributed to the nucleation effect and short-range diffusional hopping at the bulk/matrix interface [28,29]. The β grains transform into α' martensite without atoms diffusion in the form of coherent shearing under the high cooling rate [30]. Due to the rapid heating and cooling process of laser welding, non-equilibrium transformations are introduced in different zones of the welded joint which is mainly affected by the cooling rate. The β grains in the FZ grow along the temperature gradient direction in the cooling process and the parallel primary martensite rapidly nucleates and grows until meet the β grains boundary when the temperature drops below the critical temperature. In addition, the growth direction of secondary martensite is perpendicular to the primary martensite, and its nucleation and growth depend on the cooling rate. As shown in Figure 2b, the faster cooling rate limits the growth of primary martensite and promotes the nucleation of secondary martensite to form fine needle-like α' martensite. With the increase of heat input, there is enough time for the primary martensite growth and partial becomes lath-shape α' martensite with increased width. It is worth mentioning that α' martensite length is mainly limited by the size of the prior β grains [31]. The increase of heat input caused the increase of α' martensite length indicating the coarsening of the prior β grains in a certain. The zigzag α phase in HAZ is generated by bulk transformation and the change of heat input affects its internal twinning and dislocation, while it has little effect on the morphology and size.

The microstructure is the most important factor affecting the mechanical properties. α' martensite is a supersaturated solid solution and alloying elements play a role in solid solution strengthening [32]. In addition, the presence of a large number of deformation twins and stacking faults in the FZ makes it difficult for dislocations to slip [33,34] which result in the increase of the strength in the welded joint. The yield strength ($\sigma_s$) of welded joints can be expressed by the Hall–Petch formula [35,36]:

$$\sigma_s = \sigma_0 + kd^{-\frac{1}{2}} \tag{2}$$

where $\sigma_0$ is the tensile force required for dislocation movement, k is a constant related to the microstructure, and d is the average diameter of the particle size. The relationship between yield strength and hardness is shown in Equation (3)

$$\sigma_s = kH + b \tag{3}$$

where, k and b are the coefficients related to the microstructure, and H is the microhardness. Therefore, the finer grains result in the higher strength and hardness. Due to the Marangoni convection effect caused by surface tension during welding, the top of FZ is wider and the heat dissipation conditions are better than other zones of FZ resulting in its finer grains and higher microhardness (Figure 6).

The main factors affecting the corrosion resistance of welded joints are the microstructure type and grain size, among which the microstructure type dominates. The studies show that the corrosion resistance of α' martensite and zigzag α is better than that of α phase in BM in the simulated artificial saliva solution [23]. As shown in Figures 2–4, the changes of heat input affect the grain size of FZ rather than the microstructure type. The binary linear regression fitting method is used to analyze the relationship between $R_{ct}/i_{corr}$ and the length/width of α' martensite, where the length of the length of α' martensite can represent the size of the prior β grain. The fitting results are shown in Figure 11 which is in good agreement with the experimental results and the fitting formulas are as follows:

$$R_{ct} = 313,793.47826a - 4,519,010.86957b + 1,291,818.47821 \tag{4}$$

$$i_{corr} = -0.25212a + 3.65647b - 0.58349 \tag{5}$$

where $a$ is the length of $\alpha'$ martensite (µm), and $b$ is the width of $\alpha'$ martensite (µm). The absolute values of the b coefficient in the two formulas is about 14 times than that of a coefficient indicating the width of $\alpha'$ martensite has a greater effect on the corrosion resistance.

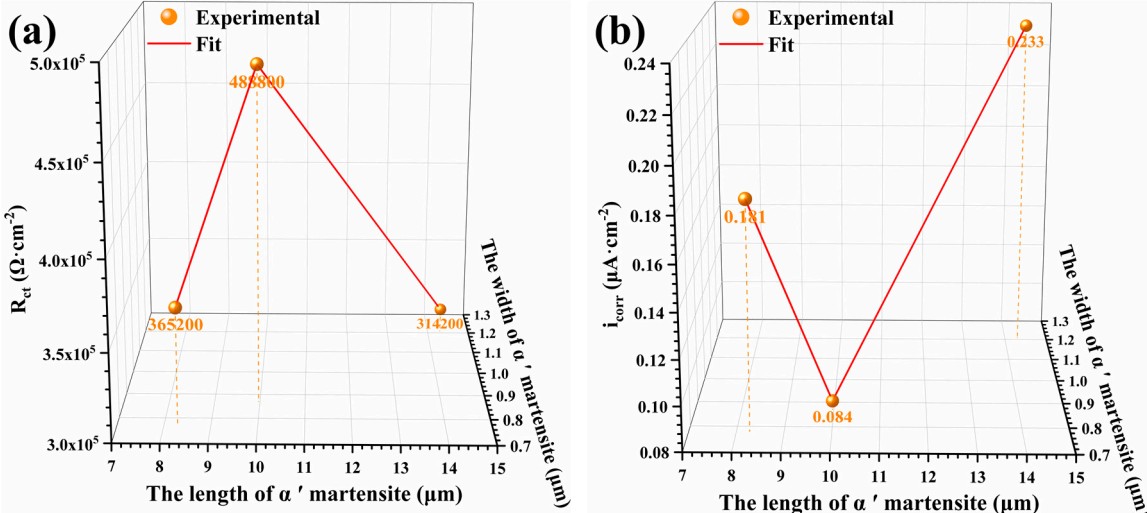

**Figure 11.** The fitting results of the relationship between (**a**) $R_{ct}$; (**b**) $i_{corr}$ and the length/width of $\alpha'$ martensite in FZ.

Studies have shown that passivation first occurs at the lattice defects sites on the titanium surface, and these defects often concentrate in the grain boundaries [37]. Alloys with finer grains have a higher density of grain boundaries, and passive films have a higher density of nucleation sites, which results in a higher proportion of passive films and lower corrosion rates. However, a more severe non-equilibrium transformation is introduced in the welded joint when the heat input is low (133.3 J/mm) resulting in a more unstable state of $\alpha'$ martensite and its preferentially dissolution in the corrosion process. In addition, the effect of the change of heat input on the corrosion resistance of HAZ is relatively small. The changes in corrosion resistance may be attributed to the formation of defects such as internal twins in the zigzag $\alpha$ phase during cooling under different heat input conditions.

## 5. Conclusions

By studying the microstructure, mechanical properties, and corrosion resistance in simulated artificial saliva solution of welded joints under different heat inputs, the following conclusions are obtained:

(1) The microstructure of FZ is needle-like $\alpha'$ martensite. The volume fraction of needle-like $\alpha'$ martensite decreases, the distribution is relatively chaotic, and the grain size increases with the increase of heat input. The microstructure of HAZ is zigzag $\alpha$ phase and the change of heat input does not significantly change its shape and size.

(2) With the increase of heat input, the elongation increases, while microhardness and tensile strength decrease. In addition, the microhardness of welded joints gradually increases from BM to FZ.

(3) The increase of heat input does not change the corrosion resistance law of each zone of the welded joint as FZ > HAZ > BM. The corrosion resistance of FZ and HAZ increased first and then decreased with the increase of heat input.

(4) The mathematical model between grain size and corrosion resistance in FZ was established by multivariate linear fitting method, it is found that the width of $\alpha'$ martensite is the main factor affecting the corrosion resistance.

**Author Contributions:** Conceptualization, W.Z.; methodology, Z.L. and W.Z.; software, Z.L.; validation, Z.L., W.Z. and K.Y.; formal analysis, Z.L.; investigation, Z.L. and N.G.; resources, W.Z.; data curation, Z.L. and K.Y.; writing—original draft preparation, Z.L.; writing—review and editing, W.Z.; visualization, Z.L.; supervision, S.G.; project administration, W.Z.; funding acquisition, W.Z. All authors have read and agreed to the published version of the manuscript.

**Funding:** The work was supported by the National Nature Science Foundation of China (No. 51805285), the Project funded by China Postdoctoral Science Foundation (2019M661016), the Key Research and Development Project of Shandong Province (2021LYXZ014), the projects of Shandong Province "Youth innovation Science and Technology Support Plan" (2021KJ026), and the Innovation Team Project of Jinan (2019GXRC035).

**Institutional Review Board Statement:** Not applicable.

**Informed Consent Statement:** Not applicable.

**Data Availability Statement:** Not applicable.

**Conflicts of Interest:** The authors declare no conflict of interest.

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
