# Peer review of "Effect of Heat Input on Microstructure and Corrosion Resistance of CP-Ti Laser Beam Welded Joints"

_metals, doi:10.3390/met12101570_

Round 1

Reviewer 1 Report

Paper reviewed: Effect of heat input on microstructure and corrosion resistance of CP-Ti laser beam welded joints

Abstract: 
FZ need to be used with full name and abbreviation in brackets - line 11

line 70-76, what is to be done is clear, however, it does not say for what application/s

It is not explicit why artificial saliva was used.

Check correct labelling of figure 2-4, it is not clear (a) FZ; b)  (a) local enlarged view; (c)HAZ

Reviewer 2 Report

Dear authors, the work is interesting, although it does not present very new data. In its current state, unfortunately, it is not suitable for printing, and requires correction. First, the image quality is very poor, some are unreadable. All images need to be improved. The microstructures in their current form do not contribute much to the job. The descriptions are rather correct, they will probably need correction after correction, but the authors think they will. Unfortunately, the conclusions are very general and, in my opinion, they cannot remain so. Please, specify.

Reviewer 3 Report

The manuscript presents the results of original researches  of the microstructure, mechanical properties and corrosion resistance  in a simulated saliva solution of welded joints of commercially pure titanium alloy TA1. Welded joints were obtained using a fiber laser welding  at various energy inputs. The results obtained by the authors indicate that the microstructure in the melting zone during laser welding changes its morphology depending on the input energy of laser radiation. It is shown that the microstructure in the zone of weld formation is a cruciform α'-martensite and lamellar α'-martensite, and in the heat-affected zone it is a zigzag α-phase of titanium. With an increase in heat input in the melting zone, the volume fraction of cruciform α'-martensite decreases and the microstructure becomes coarser, however, the microstructure in the heat-affected zone changes insignificantly.

The results of an experimental study of the corrosion resistance of a welded joint indicate that the heat input of a laser beam has a significant effect on the corrosion resistance characteristics of welded joints. An analysis of the obtained results indicates that the width of α'-martensite is the main factor affecting the corrosion resistance. The grain size distribution in the weld zone also affects the corrosion resistance.

The results obtained by the authors are of interest to a wide range of specialists working in the development and production of dental implants from alpha titanium alloys.

The quality of the manuscript could be improved.

1) It is necessary to make a correction in the text of the manuscript and clarify the abbreviations (FZ, HAZ, BM) at their first use. In the text presented, explanations are placed after repeated use of the indicated abbreviations.

2) Line 175. The statement is not formulated correctly. Figure 7 and Table 3 do not show data on the modulus of elasticity of the welded joint, but show the stress-strain curves and strength characteristics of the welded joint of the TA1 titanium alloy. Text correction required.

3) It is advisable to supplement the text in Section 3.3.3  with the formula used to fit the results and an explanation of the nsl parameters given in Table 5.

Round 2

Reviewer 2 Report

In this form the paper can be published.